# Changes in Rumen Microbiota Affect Metabolites, Immune Responses and Antioxidant Enzyme Activities of Sheep under Cold Stimulation

**DOI:** 10.3390/ani11030712

**Published:** 2021-03-05

**Authors:** Hongran Guo, Guangchen Zhou, Guangjie Tian, Yuyang Liu, Ning Dong, Linfang Li, Shijun Zhang, Haochen Chai, Yulin Chen, Yuxin Yang

**Affiliations:** College of Animal Science and Technology, Northwest A&F University, Yangling 712100, Shaanxi, China; 18392149566@163.com (H.G.); aqianli@163.com (G.Z.); 18392408568@163.com (G.T.); 18392504884@163.com (Y.L.); dongning_915@163.com (N.D.); llf151298@163.com (L.L.); caoyezhangshijun@163.com (S.Z.); haochen_cn@163.com (H.C.); chenyulin@nwsuaf.edu.cn (Y.C.)

**Keywords:** cold temperature, wind speed, Lachnospiraceae, antioxidant enzymes, cytokines

## Abstract

**Simple Summary:**

Under a cold environment, the animal’s weight is reduced and even health is affected. As we all know, microbiota is beneficial to animal health. It can produce metabolites to improve animal immunity and avoid damage. Therefore, we aimed to understand the self-protection mechanisms of sheep under cold stress. To investigate this mechanism, we designed two experiments to explore the effects of low temperature and wind speed on sheep phenotypes, rumen microbes, immune cytokines and oxidative stress. Our results identified that the sheep remained healthy in a cold environment. This may be due to the enrichment of Lachnospiraceae in the rumen. A large amount of propionate may enter into the gluconeogenesis reaction, resulting in a decrease in the content of propionate in the rumen, thereby reducing animal’s immunity. In summary, the increase of Lachnospiraceae and propionate in the rumen may help sheep live in a cold environment. Our experiments provide some direction for the healthy feeding of animals in cold environments.

**Abstract:**

Low-temperature environments can strongly affect the normal growth and health of livestock. In winter, cold weather can be accompanied by strong winds that aggravate the effects of cold on livestock. In this study, two experiments were conducted to investigate the effect of low temperature and/or wind speed on physiological indices, rumen microbiota, immune responses and oxidative stress in sheep. When sheep were exposed to cold temperature and/or stronger wind speeds, the average daily gain (ADG) decreased (*p* < 0.05), and the abundance of Lachnospiraceae was significantly higher (*p* < 0.05). The acetate and propionate contents and the proportion of propionate in the rumen also significantly reduced (*p* < 0.05). The immunoglobulin G (IgG) and TH1-related cytokines in the blood were significantly lower (*p* < 0.05). However, antioxidant enzyme contents were significantly increased and the concentration of malondialdehyde (MDA) was reduced (*p* < 0.05). In a cold environment, the abundance of Lachnospiraceae in the rumen of sheep was highly enriched, and the decreasing of propionate might be one of the factors affecting the immunity of the animals, the sheep did not suffer from oxidative damage during the experiment.

## 1. Introduction

Endothermic animals have evolved a variety of mechanisms to adapt to changes in ambient temperature. For example, mammals increase their food intake and metabolic rate to maintain homeostasis during cold winters [1]. In addition, the ruminal microbiota and mammals have a mutualistic relationship. The microbiota helps the host digest and absorb food to provide energy and small molecule nutrients for the host. This process is particularly important when animals are exposed to an extremely cold environment. In herbivores, ruminal microbes including bacteria, protozoa and fungi can produce short-chain fatty acids (SCFA) and vitamins by degrading cellulose, hemicellulose and other indigestible substances in feed. The SCFAs, such as acetate, propionate and butyrate, can be used as part of energy sources. In humans, SCFAs have been reported to enhance fatty acid oxidation, thermogenesis and immunity [2,3].

An increasing number of studies have shown that the ruminal microbiota plays a vital role in maintaining homeostasis in the host. The microbiota can be affected by genetics, diet and temperature, thereby affecting the energy balance of the host [4]. First, host genes affect the composition and function of the rumen microbiota, which in turn affect body weight and energy absorption and utilization [5]. Second, feed types such as high-fat diets and high-protein diets, different roughage: concentrate ratio, can affect the composition of the gastrointestinal bacteria [6,7,8]. Adding different levels of zinc to the piglet diets can affect the composition of the gut microbiota [9]. Additionally, feed intake also affects the microbiota. For pregnant ewes, restricted feeding can change the bacteria in the rumen, for example, to those involved in the degradation of carbohydrates and the production of short-chain fatty acids (SCFAs) [10]. Ultimately, the external environmental temperature has an impact on the microbiota. In our previous study, when goats are exposed to heat stress, the structure and composition of the rumen bacteria change significantly [11]. The cecal microbiota is affected by the cold temperature rather than feed intake, which can increase the content of SCFA and norepinephrine (NE), activate related signaling pathways, promote the expression of thermogenesis protein, and thereby increase thermogenesis [12]. Few studies have analyzed the changes in bacterial domains of the rumen under the influence of cold environments.

Disturbances in gastric-intestinal bacteria composition can cause metabolic diseases, and promote oxidative damage and gastrointestinal inflammation in both ruminants and monogastric animals [13]. The production of reactive oxygen species (ROS) is beneficial to the host through the immune response [14]. The activities of antioxidant enzymes, such as catalase (CAT) and glutathione peroxidase (GSH-PX), increased when the short-tailed field voles *Microtus agrestis* were exposed to 8 °C for 60 days [15]. This result is consistent with the increase in ROS production. In this case, the role of microbes is unknown. However, excessive production of ROS can cause oxidative stress, damage the host and cause inflammation [16]. Anti-inflammatory and pro-inflammatory factors can interact via different pathways to form a very complex inflammation regulatory network. However, few studies have demonstrated whether the changes in rumen microbiota in cold environments are related to oxidative stress and the immune response to cold temperature.

The large temperature difference between day and night during the winter and a long-term cold environment can impair the physiological balance of the sheep body, resulting in decreased immunity, slow growth and decreased production performance. The perception of a cold environment by the animal is affected by both ambient temperature and wind speed. Wind speed has a considerable influence on the degree of cold stress the animal experiences. It is suggested that the wind chill temperature (WCT) better represents cold stress [17]. The average annual wind speed in this area of China is approximately 3–4 m/s [18].

This experiment simulated the temperature and/or wind speed outside the sheep housing in a northern alpine region. We designed two experiments to evaluate the factors of temperature and wind speed, respectively. Sheep were raised under different temperatures and/or wind speeds, rumen fluid and blood were collected, and the abundance of rumen bacteria, cytokines and antioxidant enzymes were analyzed. We studied a marker of rumen bacteria using 16S rRNA gene sequencing. We found a correlation between the changes in rumen bacteria and growth characteristics, oxidative stress and immune response. This experiment explores the relations between rumen microbes and the degree of oxidative stress, as well as the regulatory function of the immune system at the molecular level in sheep under cold stress. We aim to provide a theoretical basis for sheep breeding and control under cold stress in alpine regions (Preprint) [19].

## 2. Material and Methods

### 2.1. Ethical Approval

All the animal procedures were conducted in accordance with the guidelines of the China Council on Animal Care and the Ministry of Agriculture of the People’s Republic of China. The use of animals and all experimental protocols (protocol number 100403) were authorized by the Institutional Animal Care and Use Committee of Northwest A & F University (Yangling, Shaanxi, China).

### 2.2. Experimental Animals and Design

In this experiment, the test site and the corresponding test materials were provided by Linze Experimental Station of Lanzhou University, China. The annual average wind speed in this area is about 3–4 m/s. We randomly selected 8 healthy 6-month-old crossbred ewes (small-tailed Han sheep × Hu sheep) with an average weight (BW) of 30.36 ± SE 1.68 kg; the animals were not pregnant before the experiment. The sheep were randomly and evenly divided into control groups (C) and treatment groups (LT). During the testing period, the average temperature experienced by the control group (C) was approximately 5 °C. The average temperature experienced by the treatment groups (LT) was approximately −15 °C. The experiment spanned 15 d, comprising 10 d for adaptation and 5 d for the trial period.

In the second trial, we randomly selected 4 sheep using the same requirements. We collected rumen samples and blood samples from the sheep before and after the experiment. The sheep were exposed to −15 °C and a wind treatment of 3 m/s (LW) for a 15 d adaptation period, and then exposed to −15 °C and a wind treatment of 5 m/s (HW) for 5 d. Other than the wind speed, the experimental conditions (including feeding management, test time, time of metabolic cage use, temperature, etc.) were the same between the treatment groups. The wind treatments were controlled by an industrial electric fan (fl65-1, Watson, China; maximum wind speed, 6.5 m/s). In each treatment, four fans were used to create wind in different directions to ensure a uniform wind speed. During the test period, a windproof barrier was built at the test site to avoid interference from external natural wind. The external wind speed was monitored throughout the experiment. A testo 405-v1 anemometer (testto405-v1, Testo International Trade (Shanghai) Co., Ltd., Shanghai, China) was used to measure the wind speed at different points in the sheepfold and the average wind speed was calculated. The size of the open sheep pens was 2.5 m × 2.5 m, and the distance between the fan and the + of the open sheep pens was 1.5 m. The height of the fan was 0.7 m, and the average humidity of the test site is 35%. 

### 2.3. Feeding and Management

First, the sheep were kept in closed sheep pens with an average night temperature of 5 °C (C). The sheep were kept in open sheep pens with an average night temperature of −15 °C (LT). The sheep in the C and LT groups did not receive wind stimulation. Therefore, apart from the difference in wind speed between the LT group and the LW group, the other feeding conditions were completely the same. To avoid excessive differences in wind speed among different places in the large sheepfold, two open sheep pens (2.5 m × 2.5 m) were set up in the sheepfold before the experiment began and were sterilized. The experimental sheep were fed twice daily (at 9:00 a.m. and 6:00 p.m.), some residual feed remained after each feeding, indicating that the sheep had been adequately fed. The sheep were fed a complete formula pellet feed (Gansu Yuansheng Agriculture and Animal Husbandry Technology Co., Ltd., Jinchang, Gansu, China). The feed composition and nutrient levels are shown in Appendix A. Before the test period, due to the low ambient temperature, the drinking water in the water tank completely froze for approximately 1 to 2 h each day. Therefore, the water tank was regularly monitored during the study, and any ice was removed to ensure that the sheep had access to drinking water.

### 2.4. Sampling

In each treatment group, the sheep were fed in a metabolic cage from the 3rd day to the 5th day of treatment. The sheep were weighed without feeding at 8:00 a.m. on the 3rd, 4th and 5th days of the experiment, and the average daily gain (ADG) was calculated. Additionally, all stool and urine were collected and weighed, and the daily stool and urine volumes were recorded. The collected stool and urine were then stored at −20 °C. The collected stool samples and feed samples were dried in an oven at 65 °C for 24 h. The initial moisture content was determined, and the dry stool content and dry matter intake (DMI) were calculated. We measured the DMI on the 3rd, 4th and 5th days, and the average was calculated based on the data of these three days. The dried stool samples and feed samples were crushed into a powder for testing, and the DMI and ADG were measured. On the last day of each treatment, after fasting the sheep overnight, the rumen contents were carefully pumped out, separated and stored at −80 °C. Blood samples from all sheep were collected by jugular venipuncture into a serum separator tube and immediately centrifuged at 3000× *g* for 20 min. The serum was then stored at −40 °C until the analysis of biochemical indicators.

### 2.5. Determination of Inflammatory Factors and Antioxidant Enzymes

The levels of malondialdehyde (MDA), individual antioxidant enzymes including superoxide dismutase (SOD), catalase (CAT), glutathione peroxidase (GSH-PX) and total antioxidant capacity (T-AOC) and immune factors IgG, IL-2, IL-4, IL-6 and IFN-γ levels were measured. The kits used to measure the indexes were purchased from Beijing Huaying Biotechnology Research Institute, Beijing, China.

### 2.6. Total DNA Extraction from Rumen Fluid, 16S rRNA Gene Sequencing, Data Processing and Functional Predictions

The total genomic DNA was extracted using a stool DNA kit (OMEGA Bio-Tek, Norcross, GA, USA) according to the manufacturer’s instructions. The concentration and quality of DNA were measured using a K5800 microspectrophotometer (KAIAO, Beijing, China). The V3-V4 region of the total microbial 16S r RNA gene was amplified using primers 338F (5’-ACTCCTACGGGAGGCAGCAG-3’) and 806R (5’- GGACTACHVGGGTWTCTAAT-3’) that were tailed with specific sequences and amplified genes. The amplification was performed using the following cycling conditions: 95 °C for 3 min, followed by 27 cycles of 95 °C for 30 s, 55 °C for 30 s and 72 °C for 45 s, and a final extension at 72 °C for 10 min. The products were separated on a 2% agarose gel, and the nucleotides were isolated via bead purification using the AxyPrep DNA Gel Extraction Kit (Axygen Biosciences, Union City, CA, USA). Each product was assembled in equimolar amounts and sequenced on an Illumina MiSeq platform (Illumina, San Diego, CA, USA).

The sequence data from the 16S r RNA gene were analyzed via MiSeq sequencing, quality-filtered using Trimmomatic, and merged by FLASH. The RDP classifier can quickly and accurately classify sequences into a high-order taxonomy, which can provide a range of classification structures from the domain to genus level and accurately evaluate each stage. Reads of 97% similarity were clustered into operational taxonomic units (OTUs) with ≤1% incorrect bases using UPARSE [20]. Chimeric 16S r RNA sequences were removed after CS detection [21]. QIIME was used to analyze a microbial community and graphically display the results. The functional predictions of rumen bacteria were analyzed based on PICRUSt2 using the free online platform of Majorbio Cloud Platform (www.majorbio.com, accessed on 15 June 2019).

### 2.7. Determination of Volatile Fatty Acids (VFA)

The rumen fluid was thawed on ice and centrifuged to obtain the supernatant, which was stored at 4 °C. For SCFA analysis, a solution was prepared by mixing the supernatant and butenoic acid at a ratio of 10:1 and then filtered and analyzed using gas chromatography (Agilent Technologies 7820A GC system, Santa Clara, CA, USA) as described in previous studies [11].

### 2.8. Statistical Analysis

For each 16S r RNA gene sequence, the abundances of Kyoto Encyclopedia of Genes and Genomes (KEGG) pathways were estimated by calculating the means and standard errors of the mean (SEM) using a one-way ANOVA in SPSS version 17.0 (SPSS Inc.; Chicago, IL, USA). For the bacterial community, we performed alpha and beta diversity analyses; the alpha diversity indices Simpson, Shannon and Chao were calculated, and beta diversity was explored using PLS-DA graphs. Rarefaction curves were produced in R software version 3.5.1 (R Core Team, Vienna, Austria).

The data, including ADG, antioxidant indices, immune indices and SCFA concentrations were analyzed using R software version 3.5.1. R software version 3.5.1 was used to perform Spearman correlation analysis to analyze the associations among the microbial community composition. A covariance analysis was used to determine the effect of cold stress (sheep number was included as a covariate). R was used to construct graphs. In addition, we performed a correlation network analysis. All data are presented as the mean ± SE, and values of *p* < 0.05 were considered statistically significant. The software used for the analysis is described in the Appendix A.

## 3. Results

### 3.1. Growth Performance

In these trials, the rectal temperatures of sheep were between 38 to 39 °C, and there was no significant change between groups (Appendix A). The ADG of the sheep in the LT group significantly decreased (*p* = 0.041), as well as the DMI (*p* = 0.035) (Figure 1). In the second trial, the DMI was elevated in the HW group compared with that of the LW group (*p* = 0.003). The crude fiber (CF) (*p* = 0.129) had no effect in the trials. These data indicated that the cold environment led to weight loss in the sheep.

### 3.2. Rumen Microbiota Changes under Cold Temperatures

In first trial, the microbiota community structure differed among the treatments based on a partial least squares discriminant analysis (PLS-DA) (Figure 2). Based on a Lefse analysis, *Prevotellaceae_UCG_003* was significantly enriched in the C group, and *Lachnospiraceae_XPB1014* was significantly enriched in the LT group (threshold of the LDA is 3) (Figure 2b). At the genus level, the abundance of *Prevotellaceae_UCG_003* decreased significantly in the LT group (*p* = 0.04408), and the abundance of *Lachnospiraceae_XPB1014* increased in the LT group (*p* = 0.0846). At the OTU level, the abundances of OTU103 (Family, Prevotellaceae, *p* = 0.0170), OTU493 (Family, Prevotellaceae, *p* = 0.0393) and OTU938 (Family, Lachnospiraceae, *p* = 0.0478) decreased significantly in the LT group (*p* < 0.05). The abundance of OTU498 (Family, Lachnospiraceae, *p* = 0.0064), OTU253 (Family, Prevotellaceae, *p* = 0.0258) and OTU449 (Family, Lachnospiraceae, *p* = 0.0354) increased significantly in the LT group (Figure 2c). Based on the Spearman correlation analysis, there was a significant negative correlation between the abundance of Prevotellaceae and Lachnospiraceae (Figure 2d). However, the content of volatile fatty acids did not change significantly (Table 1). These data showed a transition in abundance between Prevotellaceae and Lachnospiraceae at the family level, in response to cold temperatures.

### 3.3. Rumen Microbiota Altered with the Increase in Wind Treatment under Cold Temperatures

To verify whether the abundances of Prevotellaceae and Lachnospiraceae were associated with the cold temperature, we added wind speed to the cold environment to increase the cold stimulation. To exclude the influence of genetic background, we used four sheep and sampled the sheep at the beginning and end of the wind exposure. Based on a partial least squares discriminant analysis (PLS-DA), the microbiota community structure in the rumen differed among the treatments (Figure 3). Based on the Lefse and one-way ANOVA analyses, *Prevotellaceae_UCG_003* was significantly enriched in the LW treatment, and *Lachnospiraceae_XPB1014* was significantly enriched in the HW treatment (threshold of the LDA is 2). At the genus level, the abundance of *Prevotellaceae_UCG_003* was unchanged (*p* = 0.7323), while the abundance of *Lachnospiraceae_XPB1014* increased in the HW treatment (*p* = 0.0182) (Figure 4). The percent of propionate and the concentration of propionate and acetate decreased in the HW compared with LW treatment (Table 1). These data showed that the increase in *Lachnospiraceae_XPB1014* could be associated with the cold temperature.

### 3.4. Functional Differences in the Bacteria in a Cold Environment

The functions of the bacteria were predicted by PICRUSt2. The function of rumen microbiota changed when the wind speed increased. At KEGG level 1, environmental information processing was enriched in the HW treatment (*p* = 0.0338). At KEGG level 2, signal transduction (*p* = 0.007) and cell motility (*p* = 0.038) were increased in the HW treatment. At KEGG level 3, two-component systems (ko02020, *p* = 0.009), the HIF-1 signaling pathway (ko04066, *p* = 0.009), the AMPK signaling pathway (ko04152, *p* = 0.002) and the MAPK signaling pathway (ko04016, *p* = 0.026) were higher in the HW treatment. Flagellar assembly (ko02040, *p* = 0.061) and bacterial chemotaxis (ko02030, *p* = 0.020) also increased in the HW treatment (Table 2). These data indicated that the ruminal bacteria perceived the changes in the external environment and increased nutrient intake and utilization capacity when the cold effect was increased.

### 3.5. Changes in Inflammatory Factors and Antioxidant Enzymes

In these experiments, we found that the IgG content was significantly reduced in the HW treatment (*p* = 0.028). With regard to cytokines, IFN-γ (*p* = 0.0004), IL-2 (*p* = 0.0000) and IL-6 (*p* = 0.0006) decreased significantly in the LT and HW treatments compared with the C and LW treatments, respectively. In addition, IL-4 increased significantly in the LT and HW groups compared with the C and LW groups, respectively (*p* = 0.0068) (Figure 5).

Besides, we found that the MDA content decreased significantly in the LT group (*p* = 0.0003). The contents of antioxidant enzymes T-AOC (*p* = 0.0004), SOD (*p* = 0.0000), GSH-PX (*p* = 0.0010) and CAT (*p* = 0.0003) increased significantly in the LT group compared with those in C group (Table 3). These data showed that the immune capacity was reduced, but that the sheep may not suffer oxidative damage under cold stress.

## 4. Discussion

This trial explored the relationship between rumen microbiota and host condition in response to cold stress. Sheep were used as a model to study how animals adapt to cold. In the rumen, the abundance of Lachnospiraceae increased and the abundance of Prevotellaceae decreased. This change may have caused the decrease in acetate and propionate content. Additionally, the host experienced a reduction in the inflammatory response but did not suffer oxidative damage. The purpose of this study was to ensure the health and economic benefit of animals raised in winter by studying their physiological changes and accurately supplying nutrients.

There was significant enrichment in the Lachnospiraceae in response to cold in this study. Such changes occur regardless of the species of animal. *Lachnospiraceae uncultured* dominated in the large intestine when mice were fed cold food [22]. Cold induced *Lachnospiraceae* enrichment in the cecum of Brandt’s voles [23]. The abundance of Lachnospiraceae was inversely related to the abundance of Prevotellaceae in the rumen [24]. So there may be a decrease in Prevotellaceae when animals are subjected to cold stress.

Our results indicated that in a cold environment, the ADG of the animals decreased. When sheep were exposed to cold environments, plasma glucose and energy metabolism increased [25]. Moreover, cold stress in lambs will increase energy requirements, leading to a decrease in ADG [26]. In this case, the function of the microbiota can change to maximize the use of energy. The motility of the microbiota increased, they can move towards nutrients, ensuring a steady state of cell energy and normal proliferation. The colonization of sterile mice by *Lachnospiraceae* (strain AJ110941) led to the accumulation of liver and mesenteric fatty tissue, which benefited energy storage [27]. According to the studies, *Prevotellaceae* usually dominates the rumen and its main fermentation products are acetate and propionate [28,29]. Therefore, the decrease of Prevotellaceae can be accompanied by a decrease in the content of acetate and propionate. Propionate is a substrate of gluconeogenesis and can activate gluconeogenesis gene expression to maintain energy homeostasis [30]. However, the host may absorb a large amount of propionate for gluconeogenesis only when sheep were subjected to deeper cold stress, which will lead to a decrease in the proportion in the rumen. Propionate can promote the production of antimicrobial peptides by epithelial cells, which plays a very important role in the regulation of gastrointestinal homeostasis [31].

Other studies have shown that the immunity of animals can be reduced in a cold environment [32]. The maximum level of IgG in the blood is considered the most important immune factor [33,34]. The IgG content in plasma decreased when calves were placed in a cold (4 °C) environment for 72 to 96 h [35]. Cold stress changed the Th1/Th2 ratio and impaired the production of Th1 protective cytokines by affecting the activation and differentiation of stimulated dendritic cells [36]. Short swims in cold water can reduce the release of IL-1 and IL-6 in whole blood, and PBMC cultures are stimulated by lipopolysaccharide (LPS) [32]. The level of IL-2 decreases in the blood of mice exposed in cold temperature [37]. These observations are consistent with our findings.

Sheep exposed to cold temperatures can metabolize large quantities of nutrients to increase their heat production, which could result in oxidative damage [38]. Our preliminary judgment of whether the animal is healthy was based on the degree of oxidative damage. In the present study, the MDA content, which reflects the extent of cell damage, was decreased in the cold-exposed sheep [39]. In addition, elevated levels of antioxidant enzymes in sheep indicated oxidative stress but the host was not damaged by oxygen free radicals [40]. Overall, the sheep did not experience oxidative damage.

## 5. Conclusions

In this experiment, we concluded that the increase in Lachnospiraceae and the decrease in Prevotellaceae may have caused the decrease in ruminal acetate and propionate contents, which in turn weakened the protective effect of the immune system. However, the sheep did not suffer from oxidative damage during the experiment. We would suggest that dietary supplementation of Lachnospiraceae or SCFAs might promote adaptation to cold stress in sheep. Therefore, the molecular mechanism by which Lachnospiraceae affects the energy balance between the host and the immune response is well worth exploring.

## Figures and Tables

**Figure 1 animals-11-00712-f001:**
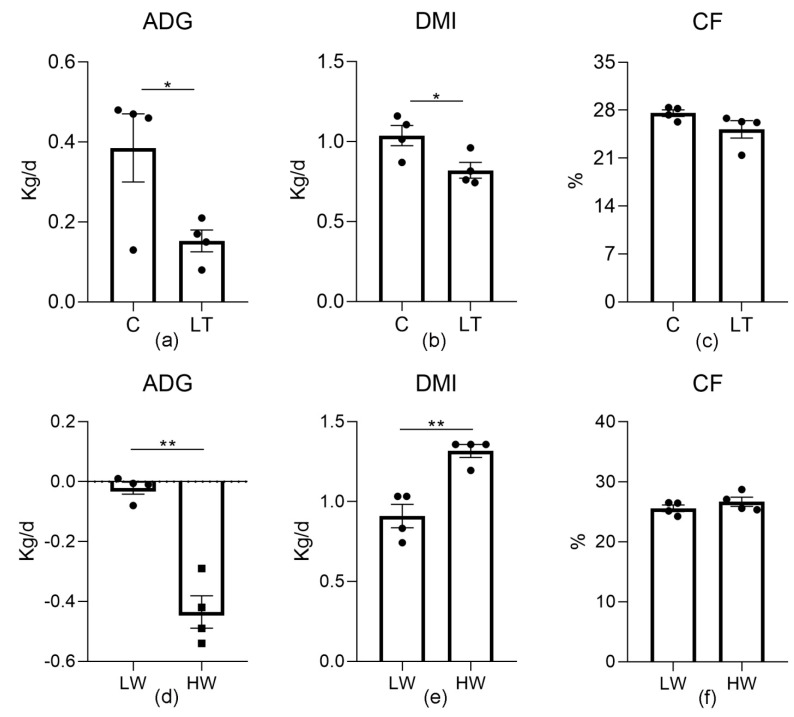
Effect of low temperature on the growth index of sheep. (**a**,**d**) Average daily gain (ADG) changes in sheep. (**b**,**e**) Dry matter intake (DMI) of each group of sheep. (**c**,**f**) Crude fiber (CF) in stool. C, sheep exposed to 5 °C, and LT, sheep exposed to −15 °C; LW, sheep exposed to −15 °C and an average wind velocity of 3 m/s, and HW, sheep exposed to −15 °C and an average wind velocity of 5 m/s. *, *p* < 0.05; **, *p* < 0.01.

**Figure 2 animals-11-00712-f002:**
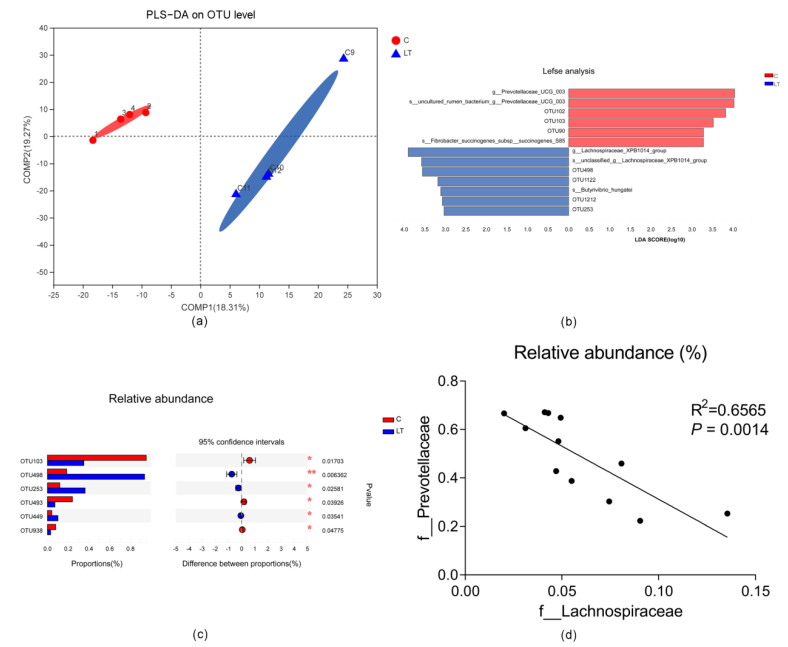
Effects of cold temperature on the microbiota of the rumen. (**a**) Partial least squares discrimination analysis (PLS-DA) of microbial diversity in the C and LT groups. (**b**) Lefse analysis of microbial diversity in the C and LT groups. (**c**) Bacterial composition at the OTU (operational taxonomic units) level. (**d**) The relationship between Lachnospiraceae and Prevotellaceae based on a Spearmen analysis. C, sheep exposed to 5 °C, and LT, sheep exposed to −15 °C. *, *p* < 0.05; **, *p* < 0.01.

**Figure 3 animals-11-00712-f003:**
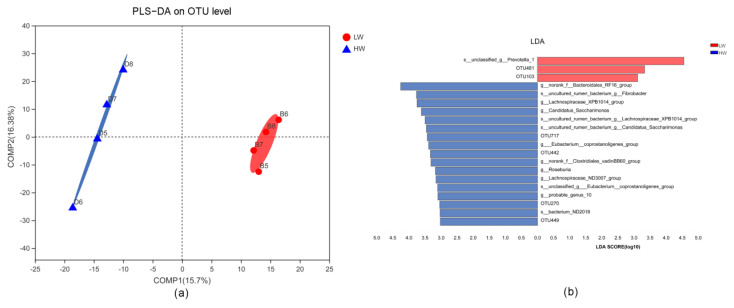
Effects of wind chill on the microbiota of the rumen. (**a**) Partial least squares discrimination analysis (PLS-DA) of microbial diversity in the LW and HW groups. (**b**) Lefse analysis of microbial diversity in the LW and HW groups. LW, sheep exposed to −15 °C and an average wind velocity of 3 m/s, and HW, sheep exposed to −15 °C and an average wind velocity of 5 m/s.

**Figure 4 animals-11-00712-f004:**
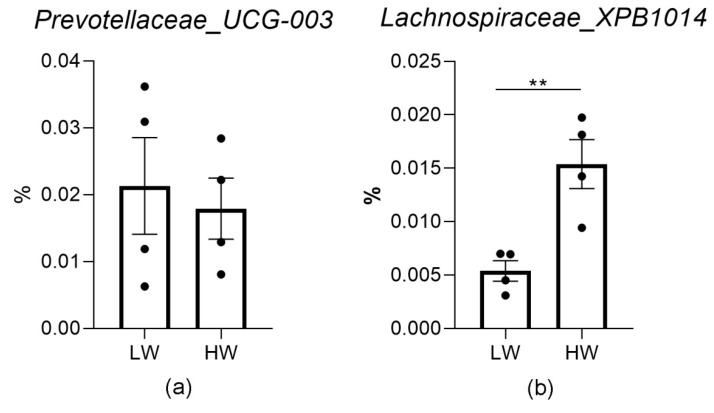
Bacterial composition at the genus level. (**a**) *Prevotellaceae_UCG-003*, (**b**) *Lachnospiraceae_XPB1014*. **, *p* < 0.01.

**Figure 5 animals-11-00712-f005:**
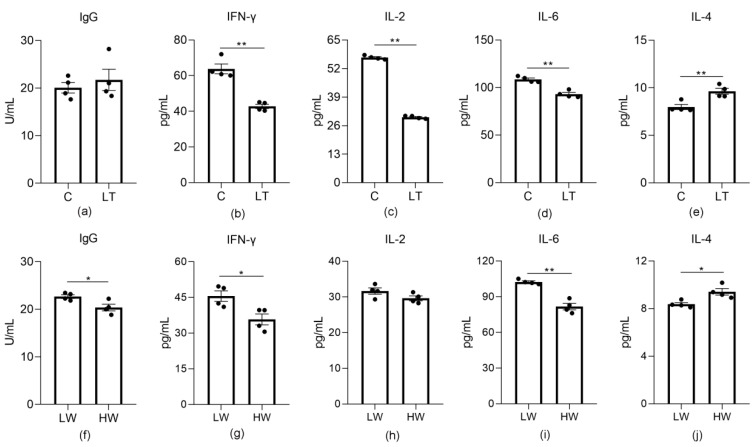
Effects of cold stress on immune factors in sheep. (**a**–**e**) Change of factors in the C and LT groups. (**a**) IgG. (**b**) IFN-γ. (**c**) IL-2. (**d**) IL-6. (**e**) IL-4. (**f**–**j**) Change of factors in the LW and HW groups. (**f**) IgG. (**g**) IFN-γ. (**h**) IL-2. (**i**) IL-6. (**j**) IL-4. *, *p* < 0.05; **, *p* < 0.01.

**Table 1 animals-11-00712-t001:** The proportion of volatile fatty acids (VFAs) in the rumen under the cold stress.

VFAs	C	LT	SEM	*p* Value	LW	HW	SEM	*p* Value
Acetate (%)	61.90	65.08	1.130	0.176	64.50	63.49	0.610	0.454
Propionate (%)	24.85	23.82	1.026	0.655	24.29	20.46	0.842	0.006
Butyrate (%)	10.47	9.026	0.600	0.259	8.504	8.745	0.382	0.778
Acetate (mM)	52.40	55.52	4.734	0.769	48.44	25.50	5.440	0.018
Propionate (mM)	21.41	21.09	2.623	0.956	18.28	8.123	2.254	0.007
Butyrate (mM)	8.797	8.209	1.106	0.813	6.506	3.474	0.843	0.064

**Table 2 animals-11-00712-t002:** The functional predictions for rumen microbes under the cold stress (%).

Pathway Level1	Pathway Level2	Pathway Level3	Description	C	LT	SEM	*p* Value	LW	HW	SEM	*p* Value
Environmental Information Processing				12.44	12.56	0.358	0.885	11.95	13.05	0.280	0.034
Environmental Information Processing	Signal transduction			8.444	8.462	0.264	0.976	11.89	13.11	0.270	0.007
Environmental Information Processing	Signal transduction	ko02020	Two-component system	12.33	12.67	0.430	0.718	11.77	13.23	0.325	0.009
Environmental Information Processing	Signal transduction	Ko04066	HIF-1 signaling pathway	12.82	12.18	0.478	0.547	11.87	13.13	0.281	0.009
Environmental Information Processing	Signal transduction	Ko04152	AMPK signaling pathway	12.32	12.68	0.563	0.774	11.52	13.48	0.408	0.002
Environmental Information Processing	Signal transduction	Ko04016	MAPK signaling pathway	13.74	11.26	1.318	0.389	11.09	13.91	0.692	0.026
Cellular Processes	Cell motility			8.043	8.673	0.746	0.706	10.81	14.19	0.868	0.038
Cellular Processes	Cell motility	Ko02040	Flagellar assembly	12.09	12.91	1.294	0.779	10.74	14.26	0.975	0.061
Cellular Processes	Cell motility	Ko02030	Bacterial chemotaxis	11.96	13.04	0.924	0.597	10.91	14.09	0.763	0.020

**Table 3 animals-11-00712-t003:** Effects of cold stress on antioxidant indices in sheep.

	C	LT	SEM	*p* Value	LW	HW	SEM	*p* Value
MDA (nmol/mL)	2.189	1.478	0.141	0.000	1.676	1.376	0.112	0.200
T-AOC (U/mL)	8.043	10.74	0.540	0.000	8.395	9.641	0.343	0.060
SOD (U/mL)	65.38	82.52	3.321	0.000	82.73	88.96	1.952	0.114
GSH-PX (U/mL)	258.2	378.4	24.55	0.001	319.3	349.1	14.93	0.356
CAT (U/mL)	17.24	25.56	1.607	0.000	20.87	21.22	0.837	0.854

## Data Availability

Raw Illumina sequencing data have been deposited in Sequence Read Archive (SRA). BioProject’s metadata are available at the following link: http://www.ncbi.nlm.nih.gov/bioproject/633534 accessed on 4 March 2021.

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
