# Peer review of "Changes in Rumen Microbiota Affect Metabolites, Immune Responses and Antioxidant Enzyme Activities of Sheep under Cold Stimulation"

_animals, 2021, doi:10.3390/ani11030712_

Round 1

Reviewer 1 Report

L15-16: “a large amount of propionate may…”
L26-27: “ the propionate content of the rumen also significantly reduced.”

L31-32: re-write “but the sheep in these trails were not oxidized”. What do you mean “sheep are, or are not, oxidized”?

L36-37: change to “ambient temperature”

L38: microbiota in the gut, on the skin, or in else place? Please clarify.

L40: “in the cold”?  what happens in other temperature conditions?

L40-41: bacteria? What about protozoa, fungi, and yeast in the rumen?

L45: again, please clarify the location where microbiota inhabits.

L54-55: In winter and summer, Tibetan sheep may consume different feeds/pastures, which can definitely affect microbiota in the rumen. Are you sure that unique microbiota composition is due to cold season only?

L58-59: in ruminants or in monogastric animals, or in both? Please clarify.

L62: “voles”?

L75-77: it is meaningless to compare the annual average of wind speed amount countries. Please delete. The local environment is more critical to animals.

L10-104: Please provide the details for how you created this temperature difference (5C vs -15C)? Those temperatures were the averages of day/night temperatures or at constant (ie. no day/night variation)?

L108-109: again, please provide the relevant positions (height, distance to sheep) of 4 fans from sheep.

2.3 feeding and management: Please clarify the experiments were carried out inside an animal house, or in field.

L126: what do you mean “582 formula feed, 120 China”?

L133-134: It appears that sheep body weight was recorded on the third and fifth day, which was used to calculated ADG. How reliable of ADG for sheep over only two days?

L140: how many days were for the DMI measurement? Please clarify.

L148: MDA is not an antioxidant enzyme. Please revise.

L192: which hormones? “growth” of the animals or the rumen microorganisms? Please clarify.

L239: change “before and after the experiment’ to ‘at the beginning and end of the wind treatment”

L265: re-word “the driving force for nutrients”. It is not clear the meaning of this expression.

L269: You cited only one reference, so not “many studies have …”. Revise this sentence.

L281: revise this sentence. “… the immune capacity was … and they not oxidized…”. You meant that “the immune capacity was not oxidized”. How is the capacity not oxidized?

L291: add “stress” after “cold”

Discussion

L290-301: Please revise the whole paragraph, making it focus on your major findings. There are not clearly logic connections between sentences.

L309-310: not sure how reliable this statement is, based on that the ADG was calculated from the body weight change over only two days.

L314-315: how comes the accumulation of the liver and mesenteric fatty tissue benefit energy absorption? Please explain.

L316-323: in your C vs LT trial, rumen propionate did not change significantly. In your LW vs HW trial, rumen propionate was reduced in the HW group. Not sure how you concluded that ‘a large quantity of propionate may enter into the gluconeogenesis pathway to provide energy for the host”. Also, are you sure the rumen tissue has considerable “immune capacity”. Do you have any reference to support this statement?

L325-327: the half-life of IgG is over 20 days. The cold treatment lasted only for 15 days in your trials, shorter than that of the IgG half-life. Therefore, are you sure the change of plasma IgG concentration reflects the real change of the IgG homeostasis in your research?

L335: Oxidation is already in presence in the body. Please revise this sentence.

L341-343: MAD is a good indicator of lipid peroxidation and one of many indicators to oxidative stress. The presence of oxidative stress is one of many assessors for the healthy status of animals, probably not critical one. Your conclusion: ‘the sheep were in a healthy state during the experiment” is mainly generated based on the indices for oxidative stress, and is not sound in accordance with your results. Please revise the conclusion.

L406-408, reference. Please provide the publishing source.  

Reviewer 2 Report

General comments

The manuscript submitted by Guo et al. reports results from a sheep experiment aiming to evaluate the effect of cold stress on rumen microbiota, immune system, and oxidative response. While many studies have focused on the impact of heat stress in ruminants, and investigated changes in rumen microbiota composition, a few literature is available up to now on the effect of cold stress. In this context, the present report is interesting for the scientific community.

The authors combined rumen microbiota analyses with 16S sequencing approach to measurements of rumen fermentative parameters, blood cytokines and blood oxidative stress markers in two separate experiments.

Although the measurements which were chosen are relevant to achieve the research objectives, the experimental design can be questioned:

  • What is the wind speed in the first trial ? indeed this is not clear whether the LT treatment is similar to the LW condition, as a wind speed of 3m/s (used in the LW condition) is an annual average reported in China (see line 75).
  • Was trial 1 performed with two non-contemporaneous groups (the big change in temperature needed two distinct periods of time?), why not the same animals as for trial 2?
  • What was the humidity level ? it is reported that for heat stress, this is combination of temperature and relative humidity which brings stressful conditions.
  • Why did the authors chose to separate into 2 trials? This brings some confusion in the way data are interpreted. It would have been much simpler to have one single trial with 3 conditions, C, LT/LW, HW. Can the authors bring justification on this choice?
  • The CF measurements with digestibility cages do not seem to bring very significant information, the authors stated that because the CF values were not different there was no change in digestibility but CF was only 15% of the diet so energy can be found in other components of the diet.

Another important concern is all that is related to VFA. In the report, only the 3 main VFAs percentage is shown, but not the concentrations, and the authors make the confusion between concentration/content and proportions. It is therefore very difficult to properly interpret the data.

Moreover, there are contradictory statements in the manuscript about the association of Lachnospiraceae abundance and propionate, which is one of the main critical points of the manuscript. In the abstract, it is stated that “In a cold environment, the abundance of Lachnospiraceae in the rumen of sheep was highly enriched, and the decreasing of propionate might be…”, in the discussion “The enrichment of Lachnospiraceae increased the propionate content”.

Finally, the 16S sequencing data are incomplete as no report on alpha diversity indexes, and no information on other taxonomic levels (phyla, genera) are provided. These data are needed to see if cold stress has any impact on bacteria diversity, or if it is possible to more deeply analyze the OTU related to Lachnospiraceae family. Also, as an inverse relationship between Lachnospiraceae and Prevotellaceae seems to be found, more discussion on Prevotellaceae should be included as this family is generally dominant in the rumen microbial community.

The results on immune function are quite consistent toward a reduced Th1 response and are well discussed with supporting literature, and the data on oxidative stress markers are also interesting and well presented.

Specific comments:

The title may not be appropriate as general health is not really affected in this work, as mentioned by the authors, animals remained healthy.

L15-16: please reformulate the sentence not to start it by And…

L17: the word “survive” is maybe too strong here.

L31: please reformulate as “sheep were not oxidized” is not very suitable

L39: helps

L41: reformulate “providing the host with nutrition”

L42: please replace “food” by “feed”, which is a more correct term for animals

L43: VFAs are definitely used as a main source of energy.

L43: ref 2 and 3 are from human studies. It would be more accurate to cut the sentence into 2: “….energy source. In humans, SCFA have been reported….(2,3).”

L45: a number….has…

L51: saccharides is not really appropriate, please modify

L52: fatty acids

L53: microbiota is singular, check throughout the manuscript that verbs are written accordingly

L51-54: the succession of sentences is not logic “feed intake affects (l49)”….”it is not affecting (l53)”

L54: could you develop a bit more how microbiota would increase thermogenesis?

L58: this sentence on probiotics is out of the topic of the manuscript and should be removed

L60: please define ROS

Material and methods:

A scheme with the two experiments would help understanding what has been done in terms of groups, measurements and period durations.

As developed above, I have some concerns about this design which definitely has at least to be better justified and explained.

A relevant measure would have been animal temperature or ruminal temperature. Was it performed?

Results:

L200: precise in the treatment group

Figure 1: ADG is in gram per day as well as DMI in general. Why is figure d in a different format than the others? What does differ between LT and LW which can explain such a difference in ADG?

As mentioned above, only VFA% is shown, not content neither concentration, so the authors should include the whole VFA data and be cautious on interpretation.

L233 Spearman

Table 2: please revise format

L269: remove reference in the results section

L270: trials

Why did the authors not measure ROS? There are several methods to analyse it.

L281: please reformulate the sentence

Discussion:

L295: very confusing here. What is reported is actually a reduced % of propionate when Lachnospiraceae are enriched, right? And, there is no proof to have a cause/effect relationship between these two parameters so the authors have to be very careful in the way they interpret the data.

L305: it is stated that Lachnospiraceae produce SCFA, which is true, but this is not what can be concluded from the results as there is no change in VFA, right?

L307: quantity

L307: ref 23 reports the effect of heat stress but there is no information on rumen microbiota. How can it be linked to the present study? It reports that when DMI is reduced, VFA concentration was decreased. Which we cannot conclude with the presented data.

L309: weight loss is only observed in HW but not in LT so it cannot be generalized to cold stress.

L316-321: as stated above, the statement here is not in accordance with the data.

L326: calves

L340-342: contradictory with was is written L26-27.

Round 2

Reviewer 1 Report

The author has addressed my comments and suggestion point-by-point. 

There are some typos and English grammatical errors, which can be corrected by the technical editor of the journal in the proof-reading.  

Reviewer 2 Report

The authors have made substantial modifications to improve their manuscript, and answered quite adequately to the comments. The manuscript is now more clear. However, as important results have been highlighted as participating to the final conclusion (ie the decrease in Prevotellaceae), there are still major issues in this revision:

  • the title : is it really appropriate to emphasize on Lachnospiraceae enrichment only? a modification should be proposed.
  • it cannot be stated that Lachnospiraceae dominated, as stated line 364 of the new version. The conclusion is overall still not clear on the link between changes in microbiota composition, changes in VFA concentrations, and the impact of propionate decrease in the rumen on energy metabolism and immune system. This last part of the manuscript should be much clearer to increase the scientific soundess of the overall paper.
  • still a few sentences to reword (l139; l226 there was no....; l266 wind exposure instead of treatment; l297 the word 'trials' was actually the good one; l318 it cannot be written that 'the enrichment increased';  and avoid to start a sentence by And... (l163, l366)
